# Uncovering narrative aging: an underlying neural mechanism compensated through spatial constructional ability
Yumeng Li[1,2], Junying Zhang[3], Xin Li [1,2] ✉ & Zhanjun Zhang[1,2] ✉

"The narrative" is a complex cognitive process that has sparked a debate on whether its features age through maintenance or decline. To address this question, we attempted to uncover the narrative aging and its underlying neural characteristics with a cross-validation based cognitive neuro-decoding statistical framework. This framework used a total of 740 healthy older participants with completed narrative and extensive neuropsychological tests and MRI scans. The results indicated that narrative comprises macro and micro structures, with the macrostructure involving complex cognitive processes more relevant to aging. For the brain functional basis, brain hub nodes contributing to macrostructure were predominantly found in the angular gyrus and medial frontal lobe, while microstructure hub nodes were located in the supramarginal gyrus and middle cingulate cortex. Moreover, networks enriched by macrostructure included the default mode network and fronto-parietal network, indicating a higher functional gradient compared to the microstructure-enriched dorsal attention network. Additionally, an interesting finding showed that macrostructure increases in spatial contribution with age, suggesting a compensatory interaction where brain regions related to spatial-constructional ability have a greater impact on macrostructure. These results, supported by neural-level validation and multimodal structural MRI, provide detailed insights into the compensatory effect in the narrative aging process.

For the vast majority of people, experience in the real world can be represented as a first-person-dominated narrative[1]. "The narrative" can be regarded as a complex process of integrating meaningful units of information from multiple structural scales[2,3]. Narrative applies various cognitive operations beneath the surface of language production to generate a framework of meaning and logic[4,5]. Thus, the maintenance of narratives will enhance individuals' perceptions of this world. Because life expectancy has risen substantially, the importance of exploring the effect of aging on "the narrative" has increased.

Although detailed neuropsychological profiles in the progression of aging demonstrate deficits in language[6,7], the features reflecting linguistic decline are still unclear. Previous studies have indicated that a potentially sensitive measure of aging progression is "the narrative", also called connected speech, which is also regarded as a marker of Alzheimer's disease at the earliest stage[8,9]. For older adults, the cohesion between their discourse units becomes weaker, and the correlation between these units and narrative themes also decreases significantly[10,11]. Moreover, measures of semantic and

lexical content, as well as syntactic complexity, were detectable from the prodromal stage in patients with Alzheimer's disease[9]. However, among the different measures of narratives, the trajectories of age-related changes even varied in a positive direction. There were no significant differences in the type-to-token ratio (TTR) or the number of different words (NDW) between young patients and elderly patients. The oral vocabulary and lexical diversity of young individuals was also greater than that of elderly individuals[12–14].

The results presented above demonstrate that substantial heterogeneity among different narrative measures and tasks is still existing. One possible explanation is that "the narrative" itself involves multiple cognitive operations which exhibit distinct patterns of aging. Thus, the current research draws upon interactive-construction model theory in order to isolate the specific operation that could capture subtle changes in aging progression. According to the theory, "the narrative" has two structures. On the one hand, the formation of basic language structures requires interconnections between linguistic elements, which can be called microstructures. On the

[1]State Key Laboratory of Cognitive Neuroscience and Learning, Beijing Normal University, Beijing, 100875, China. [2]Beijing Aging Brain Rejuvenation Initiative (BABRI) Centre, Beijing Normal University, Beijing, 100875, China. [3]Institute of Basic Research in Clinical Medicine, China Academy of Chinese Medical Sciences, Beijing, 100700, China. ✉e-mail: lixin99@bnu.edu.cn; zhang_rzs@bnu.edu.cn

other hand, the macrostructure emerges as an organizational representation of conceptual units, involving a top-down framework determining the logic of the entire narrative and reflecting the mind of the narrator[1,15,16]. These two structures correspond to two perspectives on aging in linguistic features. One perspective proposes that the aging of narrative ability exhibits language-specific characteristics. The automatic processing nature of linguistic features[17] and their independence from general cognitive resources[18,19] lead the traditional view to suggest that, in the early stages of aging, there is no significant decline in the core narrative processes[20]. However, evidence also indicates that the diminished connections at the semantic, phonological, or orthographic levels are significant factors that contribute to the adverse impacts of aging on narrative production, particularly in spoken mandarin[21,22]. Other perspective suggests that the narrative aging is primarily influenced by non-linguistic-specific cognitive operations[23–25]. The decline in processing speed among older individuals has been shown to result in prolonged latency in discourse generation[26]. Also, they exhibit reduced capacity to inhibit irrelevant information, leading to more redundant language in their narrative, lacking emphasis and coherence[27]. However, this theory is restricted in its focus on cognitive functions such as processing speed, executive function, and memory, neglecting other cognitive dimensions. The current study presents a method to evaluate macrostructure which likened to a top-down conceptual framework serving as a cognitive map for language generation. As a result, spatial construction and spatial memory abilities are included along with the classic three cognitive functions. Another alternative perspective posits that language specific cognitive processed is note limited by cognitive resources, resulting in narrative aging determined by language processing system itself[18,20]. However, aging signifies a comprehensive decline in cognitive function, guiding this study to favor the previous perspective where the associated macrostructure better portrays the development of narrative skills in the elderly.

At present, it is unclear whether these aspects of "the narrative" show different trajectories in aging. A scarcity of comprehensive research is also delving into the neural underpinnings of macrostructure and microstructure. Furthermore, previous studies have shown that cortices serve as a top-down organizational resource for narrative production via the contrast of different stimulus conditions[28–31]. These regions are susceptible to the effects of aging. Moreover, the dorsolateral prefrontal cortex has been found to play a more specific role in the narrative process among older adults than among young adults[32].

In addition, recent research has indicated that since "the narrative" is relatively complex and coherent and contains rich meaning, the higher-order cortical networks, such as the DMN and the posterior medial network (PMN), may be more widely involved in the narrative than the brain regions in which discrete random stimuli are activated[33–35]. Regions of the PMN have been shown to exhibit decreased narrative-evoked responses as a consequence of aging, and this neural process significantly predicts individual differences in episodic memory[36]. Additionally, one previous study investigated cerebral plasticity in the narrative task and described both intrahemispheric and interhemispheric reorganization in the elderly group[37]. Unfortunately, few studies have investigated the underlying neural mechanisms of narrative aging, and limited neural correlates and the role of aging in this type of neural processing have been demonstrated[38].

The purpose of the current study was to integrate a wide range of clinical indicators relevant to narratives and evaluate their diverse trajectories and cognitive contributions to aging via theoretical categorization. Another aim of this study was to establish the basis of the evolution of neural correlates around narratives in individuals with normal aging and propose a standardized framework (Fig. 1) to investigate the individual differences within individuals with cognitive-neural aging. In line with these objectives, we propose a cognitive neuro-decoding approach that associates individuals' latent age-related trajectory of narratives with brain measures. This strategy mainly relies on large scale dataset of narrative and neuropsychological tests, whole-brain resting-state functional connectivity and multimode structural MRI[39,40].

In summary, our hypothesis revolves around the different structures of narratives in aging and is divided into the following three points:

1. Macrostructure involves more complex cognitive processes than microstructure and is therefore more closely related to non-linguistic cognitive abilities. This is evidenced by the fact that the resting-state functional connectivity network supporting the cognitive process of macrostructures is located at a higher functional gradient level.
2. Since macrostructure involves more complex cognitive processes, they are more susceptible to the effects of aging. The patterns of cognitive contributions to the aging of macrostructures differ from those of microstructures due to the influence of other cognitive abilities. Similarly, these patterns are supported by the corresponding resting-state functional connectivity.
3. Finally, the substrates supporting narrative aging and its underlying neural characteristics have a corresponding structural basis involving both gray matter and white matter.

## Results

### Differences in cognitive deconstruction between macro- and microstructures

We used the "comic discourse" paradigm to record participants' narrative texts and scored the macro- and microstructure using quantitative encoding (see "Methods"). The macrostructure was classified into seven items based on the logical development of the story, while the microstructure was classified into ten items according to fundamental language elements.

Multiple cognitive abilities, including memory, executive function, attention, and language, were used as independent variables in a multiple linear regression model with narrative scores as the dependent variable. The analysis revealed that these cognitive abilities were significantly associated with macrostructure. However, among these cognitive abilities, only language-related ability (verbal fluency) was significantly associated with microstructure (Supplementary Table 4). Furthermore, Seemingly unrelated regression (SUR) analysis indicated that episodic memory ($\chi^2 = 27.29$, $p < 0.001$) and attention ability ($\chi^2 = 10.39$, $p < 0.001$) contributed significantly more to macrostructure than to microstructure in narratives (Supplementary Fig. 2) (Supplementary Table 5).

### Age-related trajectories of the narrative and cognitive contributions vary with age

With increasing age, there was a noticeable reduction in macrolevel indicators of narrative ability ($r = -0.393$, $p < 0.001$), while microlevel indicators did not significantly change with age ($r = 0.044$, $p > 0.05$). Nonlinear fitting of the aging trends in narrative ability showed that a turning point for macrolevel indicators appeared at approximately 72 years; after this point, the rate of age-related changes increased faster. However, changes in microlevel indicators remained relatively stable (Supplementary Fig. 3).

The subjects were divided into higher and lower age groups using this turning point as a cutoff, and general linear regression models were constructed for both groups (Supplementary Table 6). The results suggest that spatial constructional ability significantly contributes to macrolevel narrative ability ($\chi^2 = 4.37$, $p < 0.05$), while the contribution of episodic memory decreases ($\chi^2 = 4.03$, $p < 0.05$). Only language skills contributed significantly to microlevel narrative ability with increasing age ($\chi^2 = 5.48$, $p < 0.05$) (Supplementary Fig. 4). The results indicate that macrolevel indicators are more vulnerable to aging than microlevel indicators are. The aging of macrolevel narrative ability may become quicker at a later life stage due to the decrease in episodic memory, although spatial constructional ability can compensate to some extent. In comparison, microlevel narrative ability consistently relies on language ability. The above results indicate that macrostructure is more correlated with non-linguistic cognitive abilities than microstructure and is more sensitive to aging. Thus, our behavioral data validates hypothesis one. Next, we will further emphasize the reasonableness of hypothesis one from the perspective of brain function.

**Fig. 1 | The statistical framework of this study. a** The dominant cognitive contribution of different narrative structures was determined. **b** The CPM method was used to constrain the specific narrative edges and hub nodes. The edges linked to the hub nodes representing the higher cognitive hierarchy were validated. **c** Sliding age windows were used to investigate how the narrative-related RSFC pattern changes with age. Three methods were used to decode the cognitive representation among brain regions specific to narrative aging. **d** Structural data were used to validate the functional basis of narrative aging.

## Using the cross-validation model to identify the brain regions involved with macro- and microstructures in narratives

We used functional connectivity (edges) to predict individuals' narrative scores by applying a connectome-based predictive model (CPM)[41]; the specific statistical methods and validation procedures are shown in Supplementary Fig. 5. Briefly, after constraining the narrative-related edges with a permutation test, the edges that satisfied the fitting criteria to predict the participants' narrative performance were selected. These validated edges then served to establish a model that predicted narrative performance for the testing set.

The macrostructure of the narrative could be predicted by 554 validated edges, while the microstructure required 546 edges. The predictive accuracies of the macrostructure ($r = 0.79$, $p = 1.39 \times e^{-06}$) and

microstructure ($r = 0.62$, $p = 5.25e^{-05}$) in the testing set were both significant. The specific brain regions associated with macro- and microstructures were subsequently identified via complex network analysis. We ranked nodes according to hub value and extracted the top nodes for the macro- and microstructures. For the macrostructure, the top nodes were the middle temporal gyrus, angular gyrus, medial orbital frontal cortex and frontal pole (Fig. 2a). For the microstructure, the top nodes were the supramarginal gyrus, precuneus, superior temporal gyrus and middle cingulate cortex (Fig. 2b).

Dice similarity analysis was used to examine the neural dissociation between macro- and microstructures (Dice coefficient = 0.16, $P_{permutation} = 0.203$). The results indicated that the neural substrates of different narrative structures can be separated functionally.

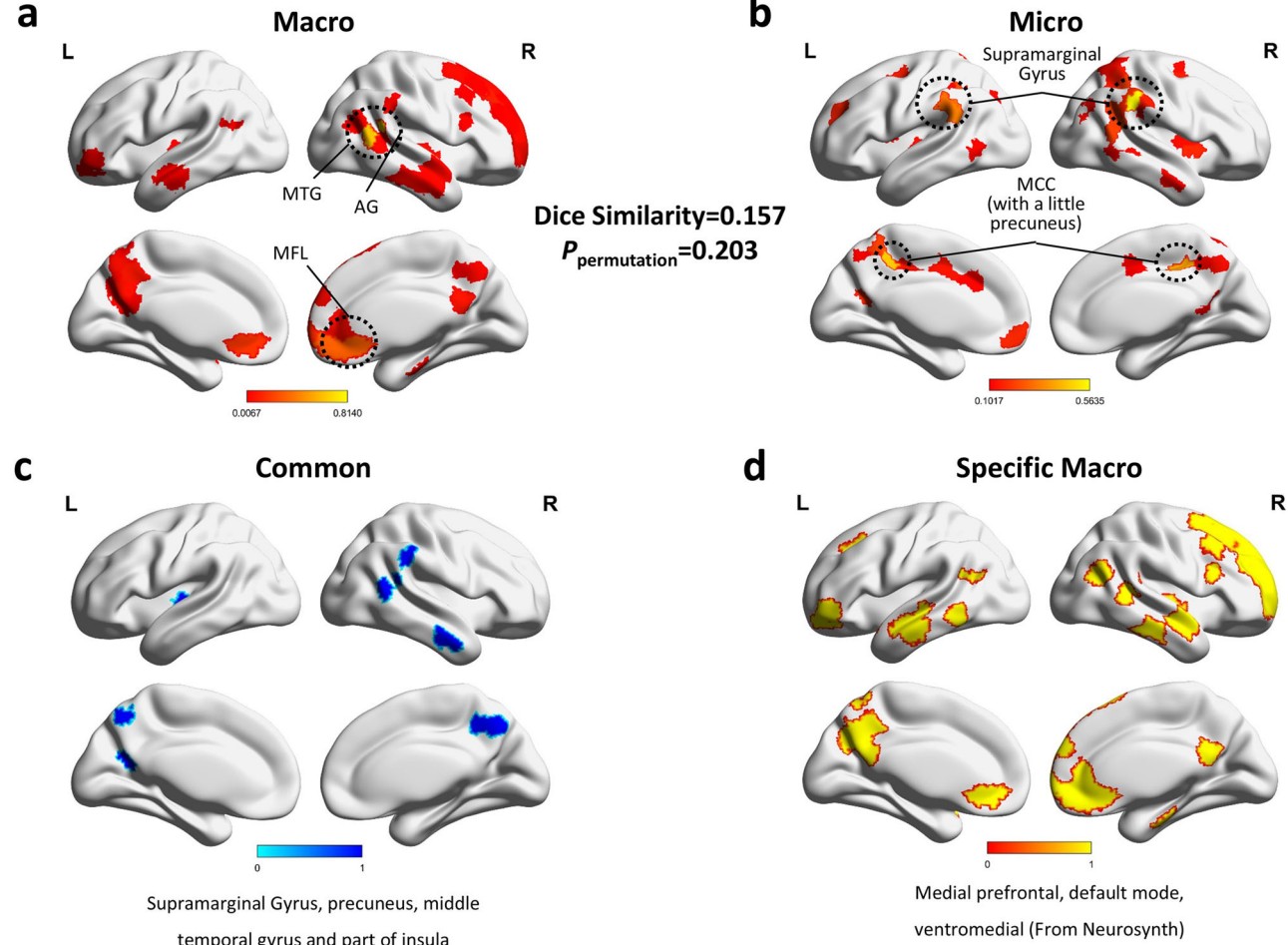

**Fig. 2 | Identification of the brain regions representing the narrative. a** The primary brain regions involved in macrostructure were the middle temporal gyrus, angular gyrus, medial orbital frontal cortex, and frontal pole. **b** The primary regions involved in microstructure were the supramarginal gyrus, precuneus, superior temporal gyrus, and middle cingulate cortex (Fig. 2b). **c** Overlap between the macro regions and micro regions, including the supramarginal gyrus, precuneus, middle temporal gyrus, and part of the insula. **d** The macrospecific map was obtained by subtracting the common map from the macro regions. The results were decoded by Neurosynth and included the medial prefrontal cortex and DMN-related areas.

We also performed conjunction analysis to reveal the overlapping brain regions between macro- and microstructures. The overlapping regions were the supramarginal gyrus, precuneus, middle temporal gyrus and part of the insula (Fig. 2c). Although these brain regions all contribute to narrative ability, the degree of importance to different narrative structures varies. For instance, the MTG serves as a hub area in the brain network of macrostructures, while the supramarginal gyrus functions as the hub area for the microstructure. In addition to the shared brain regions, brain regions specific to macrostructures included the medial prefrontal cortex and DMN-related areas compared to microstructure-correlated brain regions (Fig. 2d).

**Investigating the higher functional hierarchy of macrostructures**
The behavioral data suggest that macrostructures hold greater significance for higher-level cognitive functions than microstructures. Therefore, we examined the distribution of contributing edges that support narrative processing, aiming to identify the networks in which these edges were mainly organized.

**The enrichment of hub nodes for macro- and microstructures**. We further investigated the distribution of the contributing edges that were linked to the macro- or microstructure hub nodes[42] (see "Methods"). As shown in Fig. 3, for the macrostructure, the contributing edges that were

linked to the identified 10 nodes were mainly distributed in the FPN ($2.34 \times$ enrichment). Moreover, for specific brain regions related to macrostructures, such as the middle temporal gyrus (MTG), angular gyrus (AG) and medial frontal lobe (MFL), the contributing edges were enriched mainly in the DMN and FPN ($3.26 \times$ enrichment and $4.13 \times$ enrichment) (Fig. 3b). For the microstructure, the contributing edges that were linked to the identified 10 nodes were mainly distributed in the DAN ($1.40 \times$ enrichment) (Fig. 3a). However, for specific brain regions related to the microstructure, such as the supramarginal gyrus, middle cingulate gyrus, and precuneus, the contributing edges were enriched mainly in the DAN and VAN ($3.82 \times$ enrichment and $3.58 \times$ enrichment) (Fig. 3c).

**The enrichment of community patterns for macro- and micro-structures**. The enrichment fold of the community was calculated as described for the hub nodes, that is, the actual number of observed nodes within a specific network (such as the DMN) divided by the expected number of nodes. To compare the contributions of various networks to the communities, we employed a permutation test to calculate the significance of network enrichment scores within the communities (see "Methods").

The results indicated that the network enrichment patterns of the main communities differed between macro- and microstructures. The macrostructure was more dependent on the FPN and DMN (enrichment$_{\text{FPN}}$ = 1.19,

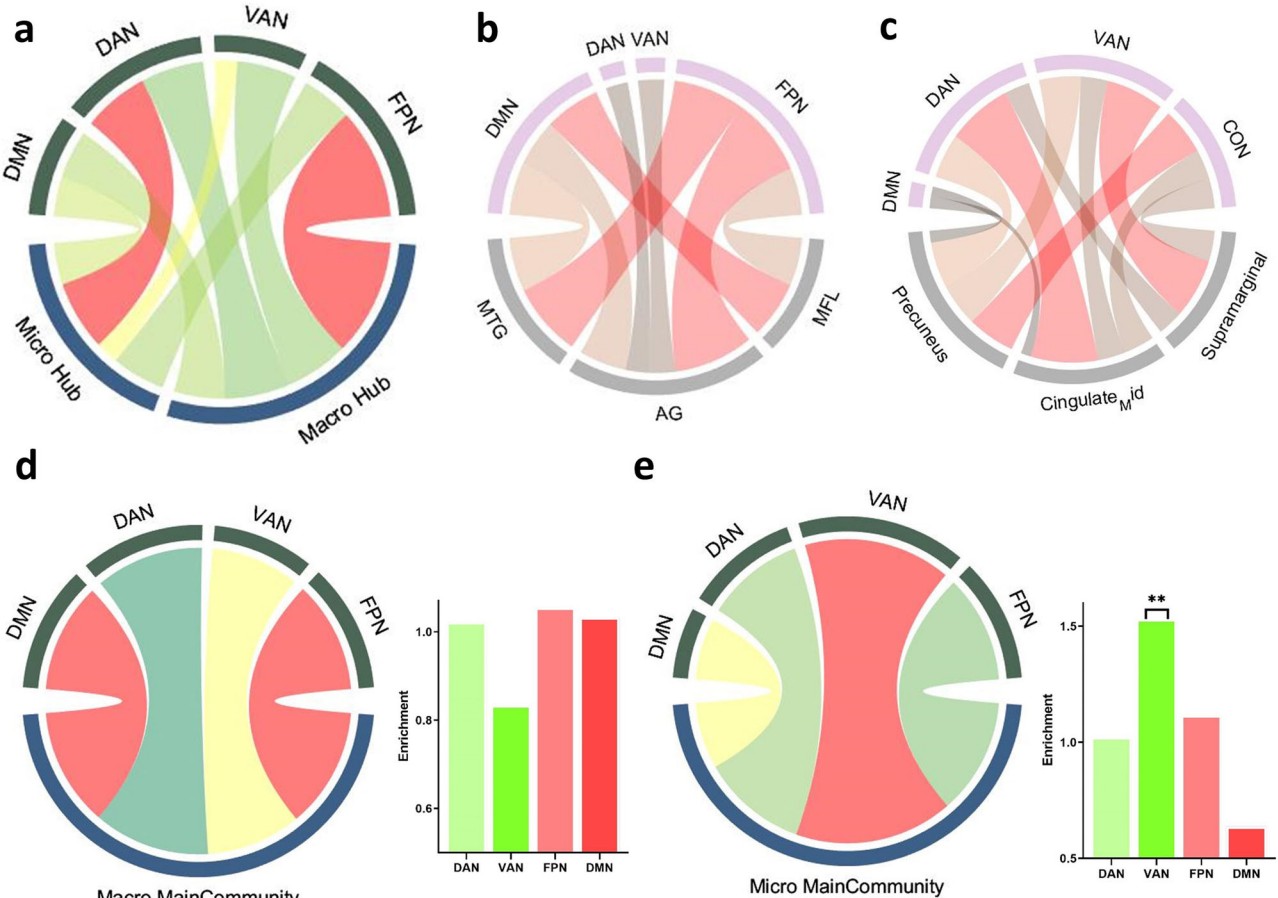

**Fig. 3 | The enrichment patterns of contributing hub nodes and communities.**
**a** The contributing edges that were linked to hub nodes of the macrostructure were mainly distributed in the FPN. The contributing edges that were linked to the hub nodes of the microstructure were mainly distributed in the DAN. **b** The contributing edges of specific brain regions for macrostructures, such as the MTG, AG, and MFL, were mainly enriched in the DMN and FPN. **c** The contributing edges of specific regions for microstructures, such as the supramarginal gyrus, middle cingulate gyrus and precuneus, were mainly enriched in the DAN and VAN. **d** The enrichment pattern of the main community of macrostructures was more dependent on the FPN and DMN. **e** The enrichment pattern of the main community of microstructures was more dependent on the VAN.

$P_{permutation} = 0.06$; enrichment$_{DMN} = 1.05$, $P_{permutation} = 0.09$) (Fig. 4d), while the microstructure was more dependent on the VAN (enrichment$_{VAN} = 1.53$, $P_{permutation} = 0.01$; enrichment$_{FPN} = 1.12$, $P_{permutation} = 0.07$) (Fig. 3e). However, the network enrichment value of the macrostructure for the main community did not reach significance, indicating that nodes from various networks are more likely to cooperate with each other to maintain the cognitive processing of the macrostructure. Therefore, the above results regarding the DMN and FPN were mainly enriched for the hub nodes and communities of the macrostructure, which demonstrated that, compared to the microstructure with more enrichment of the DAN and VAN, the macrostructure operates at a greater cognitive level.

Integrating the results of 2.3 and 2.4, we found that both the neural substrates characterizing macrostructure and the enrichment of brain networks are situated at higher functional hierarchy, demonstrating that macrostructure involves more complex cognitive processes compared to microstructure. We have verified the hypothesis one, which states that "macrostructure involves more complex cognitive processes compared to microstructure."

### Using the RSFC pattern to explain the cognitive contribution of narrative aging

The behavioral data support the second hypothesis that macrostructures are more sensitive to aging than microstructures due to the complex cognitive processes involved in episodic memory and spatial constructional abilities.

We used hub nodes from macro- and microstructures to construct the FC matrix, which was correlated with narrative performance to generate the narrative-related RSFC pattern. Then, the participants were divided into continuous age windows, and the RSFC in each window was correlated with age to determine the aging tendency of each edge to contribute to the narrative structure. The results showed that the narrative-related functional connectivity matrix exhibited both positive and negative changes as people aged (Fig. 4). The narrative-related neural substrates exhibited an age-related compensatory effect to represent the process of narrative aging; that is, the RSFC pattern of positive brain areas better represented narrative ability during aging, while the contribution of negative brain areas decreased.

### Decoding the specific function of regions contributing to narrative aging with three mutually validating methods

To further explain the change in the RSFC pattern during narrative aging and to verify similar compensatory effects in the behavioral results, we examined the contribution of increased spatial constructional ability to macrostructures and decreased episodic memory as people aged. We used three methods to determine which cognitive function is represented by the specific brain regions with convertible RSFC patterns during narrative aging (Fig. 5).

First, we constructed several models for predicting different cognitive functions using the RSFC patterns of specific brain regions. If a

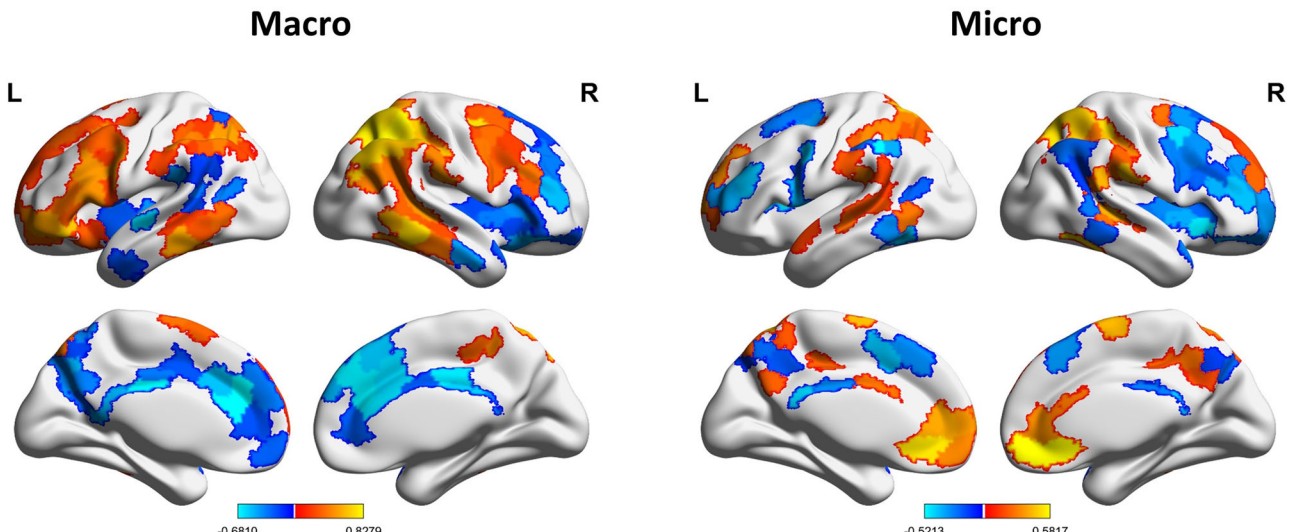

## Macro

## Micro

**Fig. 4 | An age-related compensatory effect represents the process of narrative aging.** We correlated the RSFC matrix with age in each time window to determine the aging tendency of each edge to contribute to the narrative structure. The narrative-related neural substrates exhibited an age-related compensatory effect

to represent the process of narrative aging; that is, the RSFC pattern of positive brain areas could better represent narrative ability during aging, while the contribution of negative brain areas decreased.

model performed well, the corresponding brain region was considered to represent the cognitive function. In terms of macrostructure, the RSFC pattern of positive brain regions could predict spatial constructional ability only ($r = 0.42$, $P_{permutation} = 0.015$), while that of negative brain regions could predict executive function and episodic memory ($r = 0.38$, $P_{permutation} = 0.03$; $r = 0.39$, $P_{permutation} = 0.02$). However, for the microstructure, the RSFC pattern of positive brain regions could not predict any cognitive function, while that of negative brain regions could predict verbal fluency and episodic memory ($r = 0.47$, $P_{permutation} = 0.006$; $r = 0.62$, $P_{permutation} = 1.04 \times e^{-04}$). The detailed fitting results for the abovementioned edges and different cognitive functions are presented in Supplementary Figs. 7 and 8.

Second, we used meta-analytic data from Neurosynth (https://www.neurosynth.org) to decode the specific regions. For the macrostructure, the decoding results for the positive regions included the intraparietal sulcus, posterior parietal cortex, spatial cortex, and attention and spatial constructional cortex, while the decoding results for the negative regions included the anterior cingulate, dorsal anterior cortex, medial prefrontal lobe, response inhibition, and memory. For the microstructure, the decoding results of the positive regions included the posterior cingulate, ventral medial, precuneus, default, and visual word regions, while the decoding results of the negative regions included the inferior frontal, language, phonological, sentence, and word regions.

Third, we used the method CPMto identify the hub nodes associated with different cognitive functions. These nodes were used to calculate Dice similarity coefficients with the specific regions of narrative aging. The higher the Dice similarity coefficient is, the more the specific brain regions represent corresponding cognitive abilities. The results indicated that the overlap between hub nodes and positive regions of macrostructure was significant for episodic memory (Dice coefficient = 0.15, $P_{permutation} = 7.8 \times 10^{-04}$), executive function (Dice coefficient = 0.28, $P_{permutation} = 8.4 \times 10^{-04}$) and spatial constructional ability (Dice coefficient = 0.31, $P_{permutation} < 1 \times 10^{-04}$), while verbal fluency was not significant (Dice coefficient = 0.29, $P_{permutation} = 0.49$). The hub nodes representing spatial constructional ability had the most significant overlap with the positive regions for macrostructure.

Additionally, the overlap between hub nodes and negative regions for macrostructure was significant for episodic memory (Dice coefficient = 0.37, $P_{permutation} < 1 \times 10^{-04}$), executive function (Dice coefficient = 0.15,

$P_{permutation} = 0.01$) and spatial constructional ability (Dice coefficient = 0.20, $P_{permutation} = 8.2 \times 10^{-04}$), while verbal fluency was not significant (Dice coefficient = 0.22, $P_{permutation} = 0.40$). The hub nodes representing episodic memory had the most significant overlap with negative regions for macrostructure.

Finally, we combined these three methods to identify the cognitive function that is most appropriately represented by specific brain regions (Supplementary Table 3). With regard to the macrostructure, spatial constructional abilities offer the most compensation for the aging process, while the contributions of episodic memory and executive functions diminish. In terms of microstructure, compensated or declined contributions are intricately linked to language-related abilities.

Integrating the results 2.5 and 2.6, we have verified hypothesis 2 which suggests that "due to its more complex cognitive processes, the mechanisms underlying aging effects on macrostructure differ significantly from microstructure," we found that certain brain regions play an increased role in the aging process of macrostructure (Fig. 4). To decode the specific cognitive function associated with these brain regions, we utilized three methods depicted in Fig. 5 for decoding to achieve cross-validation. Ultimately, we discovered that these brain regions correspond to spatial construction abilities.

### Investigating the structural basis of cognitive-neural mechanisms of narrative aging

By exploring the RSFC pattern associated with the aging of narrative ability, we found an underlying compensatory effect of spatial construction ability to macrostructure. However, based on the results in Fig. 4, compensatory regions also exist at the microstructure as well. To further provide a structural basis for this effect and verify the hypothesis 3, we verified the observation using both gray matter and white matter data. We theorized that the specific brain regions compensate during the narrative process due to their structural features that support this function, which is demonstrated through two results. First, the coordinated changes in gray matter structure between narrative hub nodes and compensatory brain regions will be more significant. Second, the strength of white matter connectivity between hub nodes and specific brain regions can predict changes in functional connectivity patterns associated with aging (Fig. 6).

The structural covariance network was used to calculate the coordinated change in gray matter volume (GMV) between hub

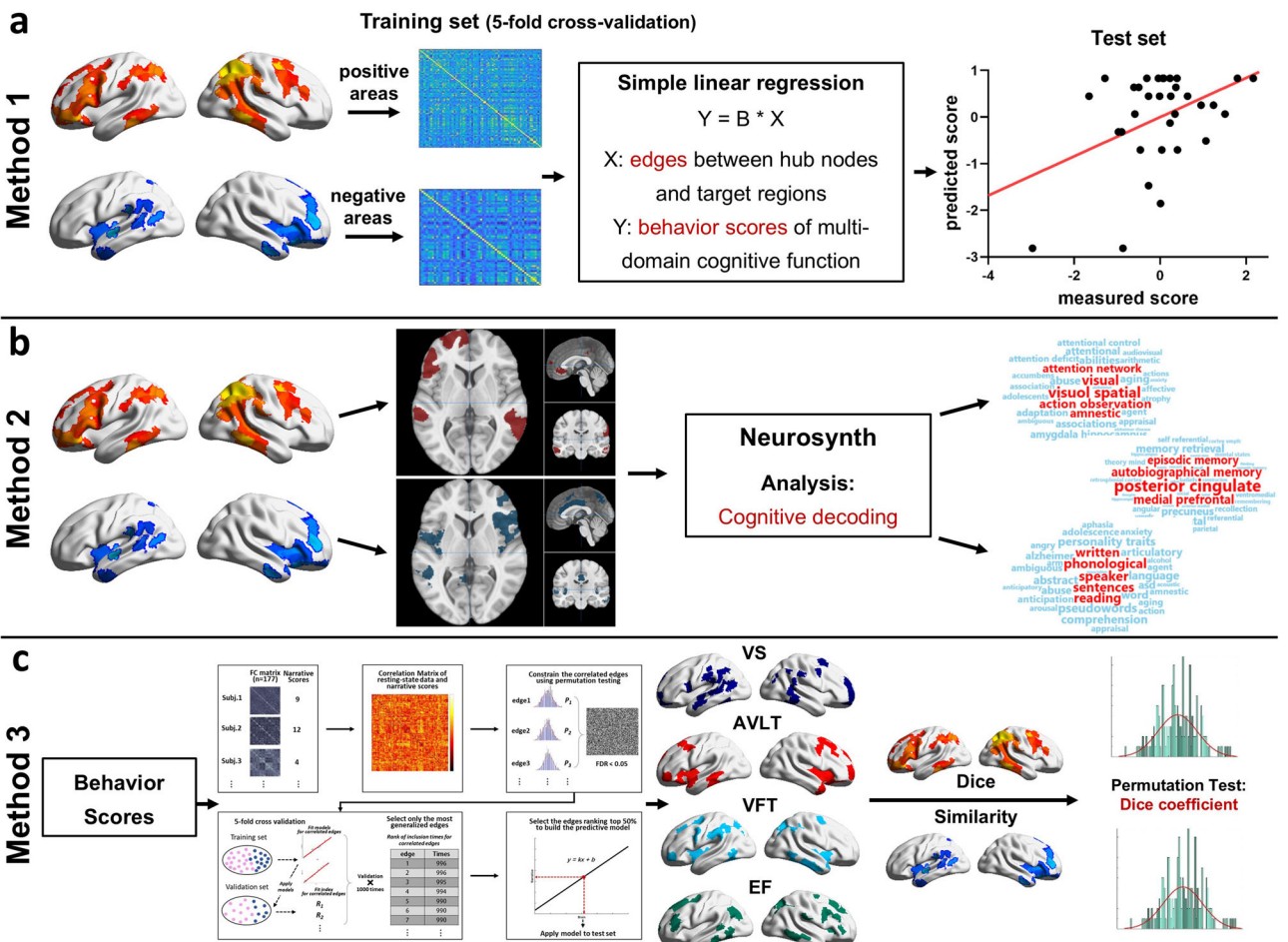

**Fig. 5 | The schematic diagram of three methods to determine the cognitive domain represented by specific RSFC patterns during narrative aging.** We used three methods to determine which cognitive function was represented by the specific brain regions with convertible RSFC patterns during narrative aging. **a** Method 1: We constructed several models predicting different cognitive functions using the RSFC patterns of specific brain regions. If a model performed well, the corresponding brain region was considered to represent cognitive function. **b** Method 2: we used meta-analytic data from Neurosynth (https://www.neurosynth.org) to decode the specific regions. **c** Method 3: We used the method in Fig. 1 to identify the hub nodes related to different cognitive functions. These nodes were used to calculate Dice similarity coefficients with the specific regions of narrative aging. The higher the Dice similarity coefficient is, the more the specific brain regions represent corresponding cognitive abilities.

nodes and specific regions (see "Methods"). The results indicate that hub nodes and compensatory regions of macrostructures exhibit more coordinated changes in gray matter volume (GMV) with age. Compared to those in the declining regions, the compensatory regions with coordinated changes were more numerous and exhibited greater significance ($Ms = 0.11$, $P_{permutation} = 0.03$). However, the related changes in the hub nodes and compensatory regions for microstructures were not significantly greater than those for the declining regions ($Ms = 0.02$, $P_{permutation} = 0.30$).

Additionally, the linear regression model using the white matter network to predict the RSFC pattern also validated the compensatory effect of specific brain regions related to narrative aging (see "Methods"). The results indicate that the white matter connectivity strengths between macrostructure hub nodes and specific brain regions (including compensatory regions and declining regions) can successfully predict age-related changes characterizing the macrostructure ($r = 0.32$, $p = 1.26 \times e^{-04}$). In other words, the stronger the compensatory effect on the macrostructure is, the greater the benefit from the stronger white matter connectivity between the macrostructure hub nodes and compensatory brain regions. However, for microstructure, the performance of the predictive model was not adequate ($r = 0.13$, $p = 0.18$).

## Discussion

We established distinct macro- and microstructures using a wide range of narrative-related indicators that are theoretically separable. Through the integration of large-scale multimodal data from brain imaging and comprehensive neuropsychological tests, we also investigated the cognitive mechanisms and neural substrates associated with narrative ability and its underlying compensated effect in aging. Our results revealed a mechanism in which macrostructures display an age-related increase in spatial contribution coupled with a decrease in episodic memory contribution. This mechanism was verified at the neural level, and the associated RSFC pattern demonstrated a structural relationship.

We obtained convergent evidence from the behavioral, functional, and structural data supporting the neural substrates of narrative components, demonstrating the complex cognitive contributions across different narrative levels. The approach used in this study provides a robust and stable statistical framework for investigating a wide range of cognitive functions. Consequently, this study has significantly contributed to our understanding of the cognitive contributions to narrative and their changes with aging, as well as their neural underpinnings. Furthermore, this study provides insight into related theories and has potential implications for the development and clinical application of innovative neuropsychological tests.

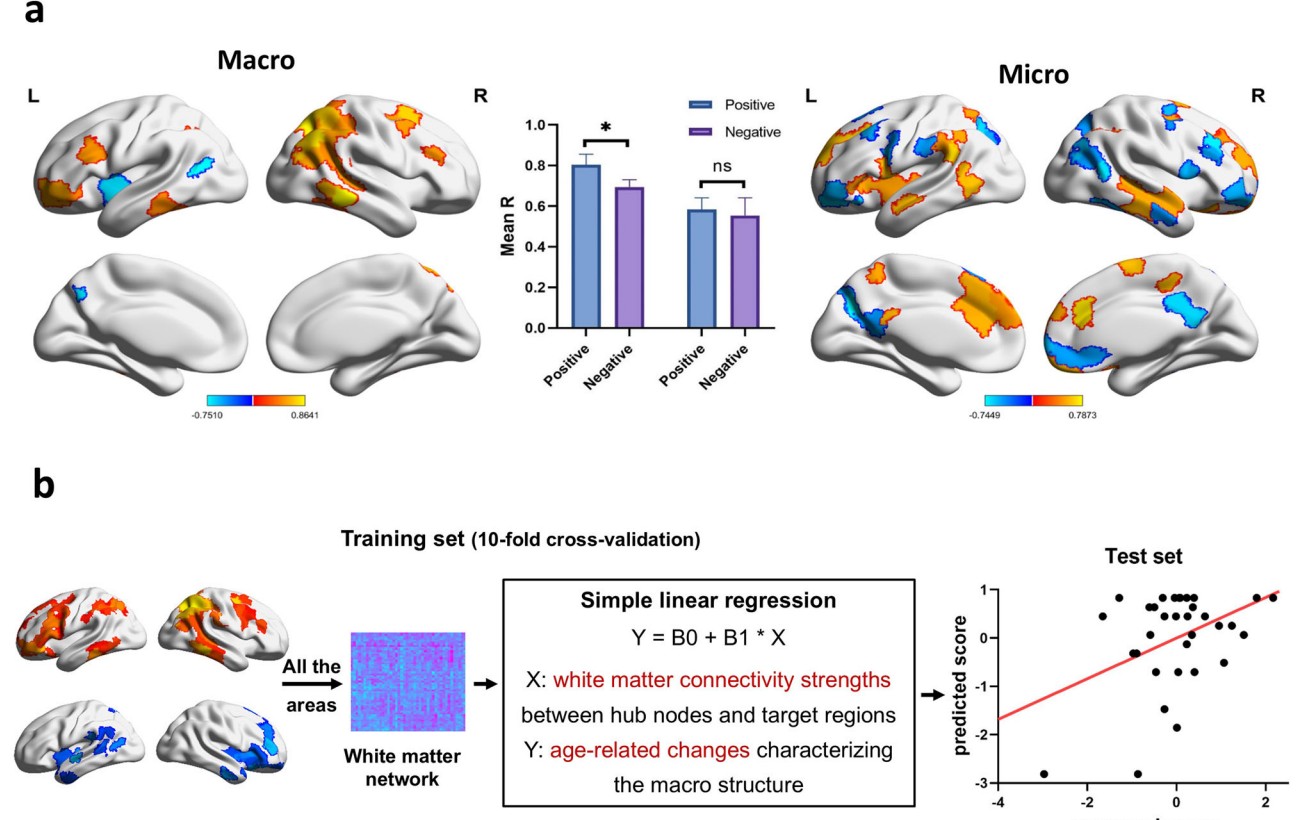

**Fig. 6 | The brain structural basis of the cognitive-neural mechanisms of narrative aging. a** The structural covariance network was used to calculate the coordinated change in GMV between hub nodes and specific regions, indicating that the coordinated changes in gray matter structure between narrative hub nodes and compensatory brain regions are more significant. **b** The linear regression model using the white matter network to predict the RSFC pattern also validated the compensatory effect of specific brain regions related to narrative aging, and the strength of white matter connectivity between hub nodes and specific brain regions can predict changes in functional connectivity patterns associated with aging.

The central focus of this study revolves around the macro- and microstructure of narratives. To distinguish these two components, previous research has used the strategy of recording young adults' brain activity when performing single narrative tasks and using conditional contrasts to identify the component associated with higher-level cognitive processes[28,30]. The activated brain regions included the inferior frontal lobe, which overlaps with the medial frontal lobe identified in this study. The inferior MFLs are also key regions for distinguishing macro- and microstructures, implying that this region plays a vital role in top-down organization for maintaining the theme and logical flow of narrative processes rather than supporting basic linguistic units[43,44]. In addition to the MFL, the angular gyrus (AG) has been overlooked in previous studies on narrative discourse, despite its involvement in various high-level cognitive functions, particularly episodic memory and semantic memory[45]. The AG plays a role in integrating and retrieving the supramodal experience, such as by integrating individual concepts into larger groups, which is consistent with the concept of schema[46,47]. Within the macrostructure, the AG is suggested to engage in integrative operations generating the conceptual frame (schema) crucial for organizational representations between conceptual units. The middle temporal gyrus also serves as one of the hub nodes contributing to the macrostructure. A previous study using near-infrared spectroscopy revealed significant activation in the left middle temporal gyrus (MTG) among older adults, potentially indicating its involvement in the semantic interpretation of narrative text[32]. However, our findings suggest that the right middle temporal gyrus could serve as the neural basis of the macrostructure. This finding is reasonable because the right MTG plays an important role in semantic violation tasks, which requires the detection of conflict arising from semantic anomalies based on the context[48,49]. Hence, the MTG appears

to engage in not only the representation of the propositional context within the macrostructure but also the semantic interpretation within the microstructure, which tends to be responsible for the intermediate transition during the narrative process. However, other studies have shown that the ability to interpret at the abstract level is more likely to be dependent on a ventral pathway linking the right MTG with the anterior inferior frontal lobe[50]. The interaction of macro hub nodes may increase the involvement of the MTG.

In contrast, the hub node that contributed most to the microstructure was the supramarginal gyrus. The supramarginal gyrus is regarded as a particularly important region in phonological processing, which is part of the initial stage of the microstructure according to the interactive-construction model[51–53]. However, the middle cingulate cortex (MCC) is involved in the recognition of smaller, meaningful events among narrative speech, which helps to establish local conceptual propositions compared to the global conceptual frame of macrostructures[54]. Moreover, the microstructure hub node is also attached to a small region of the precuneus. According to a previous study, the precuneus is a relatively high-order region with responses that are strongly contextually modulated[55,56]. Thus, the MCC-precuneus hub node can also be regarded as an intermediate region involved in the transition to the macrostructure. In summary, each region identified to represent narrative ability in the present study could contribute to a potential subnetwork that can be used to construct the neural mappings involved in the detailed process of the interactive construction model, which provides the valid logic and qualities of narrative production (Fig. 7).

Based on the hub nodes, we also analyzed the enrichment patterns of macro- and microstructures. Regardless of the hub nodes or the main

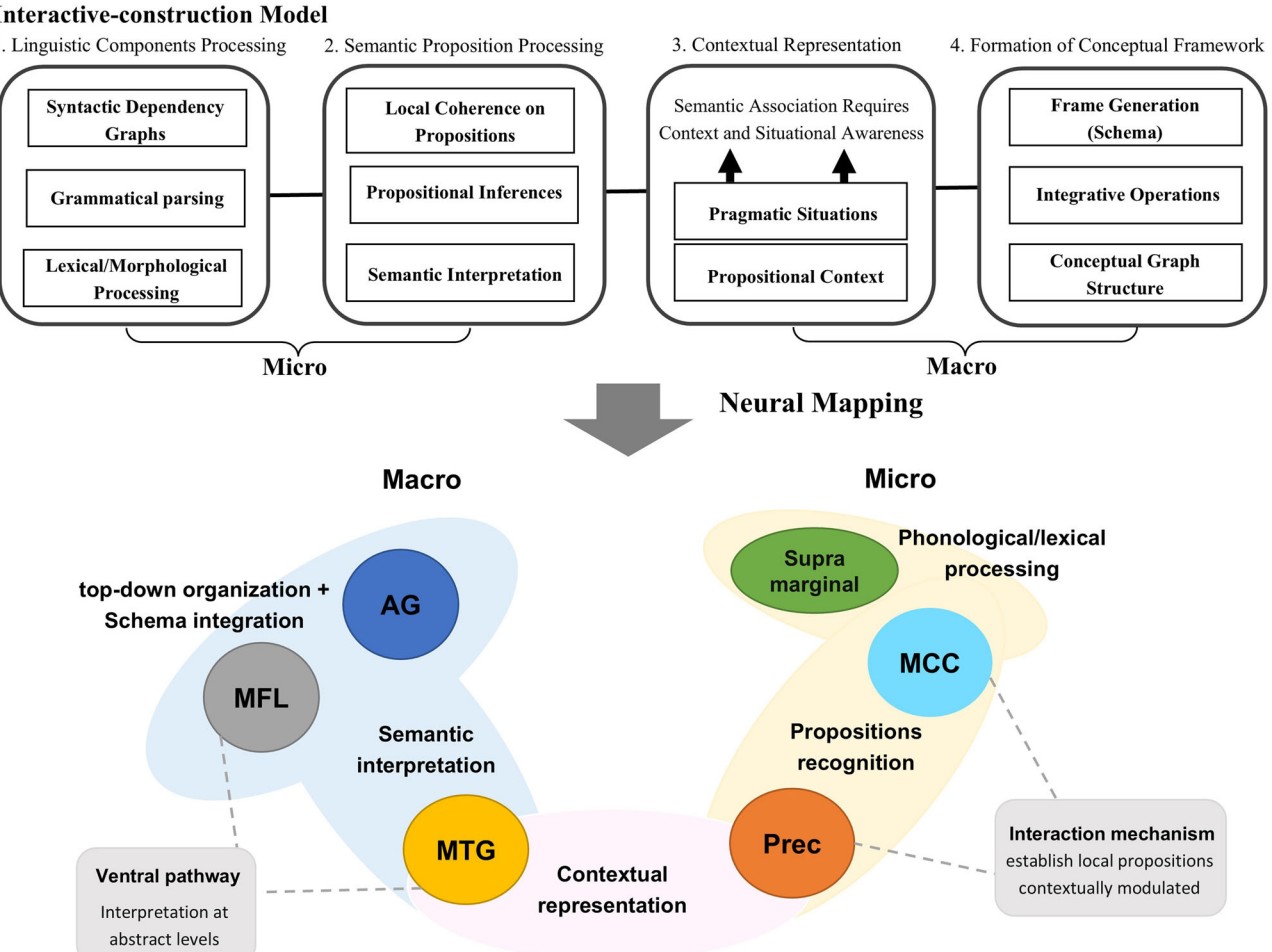

**Fig. 7 | Integrated diagram of the neural mappings of two narrative components.** The detailed processes of macro- and micronarrative are composed of four modules and their associated elements: (1) linguistic component processing; (2) semantic proposition processing; (3) contextual representation; and (4) formation conceptual framework. The neural mapping of these processes is depicted as region groupings highlighted in light yellow and blue (representing the micro- and macrostructures of the narrative). The pink groupings indicate the processes of contextual representations that relate to both micro- and macrostructures.

community, the related contributing edges were enriched in a similar pattern. For the macrostructure, the majority of contributing edges were linked to the fronto-parietal network (FPN) and default mode network (DMN). Serving as a key control system, the FPN supports context-dependent narrative operation and was demonstrated to play a role in resolving semantic ambiguity when appropriate contextual cues are lacking[57,58]. In addition, the FPN is suggested to maintain the coherence and causal relationships that encourage plot narratives, which are considered to fit well with the process of semantic connection according to the context of the macrostructure[59,60]. The DMN is the highest level in the functional hierarchy of the human brain; its contribution to various cognitive functions has been widely studied, and the narrative is no exception. Like the FPN, the DMN is also sensitive to contextual information[61]. However, the DMN can process implicit context, which suggests that the DMN is more adept at information integration and forming corresponding experiences or schemas[62]. In addition, previous studies have indicated that the DMN plays a strong role in the underlying processing of a coherent narrative, and this processing is particularly related to the encoding and comprehension of conceptual structures among content[63]. For example, the sustained activation of the DMN within its components (especially the angular gyrus) results in integrative operations and frame generation according to the structural characteristics of the narrative being processed[5,33,64,65]. Together with our results, these studies suggested that the FPN and DMN, as networks of high-level regions, are responsible for the cognitive processing of macrostructures. The FPN is involved in the construction of semantic connections in the propositional context, and the DMN is involved in information integration and the formation of a top-down conceptual framework.

For the microstructure, the majority of contributing edges were linked to the ventral attention network (VAN) and dorsal attention network (DAN). The VAN was demonstrated to highly overlap with the language network and mirror the language network in the right hemisphere of the brain[66]. Some of the constituent regions of the VAN and DAN are also the key nodes contributing to the network that is specific to the reading function[67]. This has also resulted in better reading performance when functional connectivity within the VAN is greater[68]. More specifically, previous studies have indicated that various brain regions involved in the VAN are responsible for phonological processing, syntactic graphs and semantic interpretation, which are suggested to activate the phonological, syntactic and semantic components of spoken words using executive or selective attention[69–71].

One notable finding from our study is that brain regions associated with spatial constructional ability make a greater contribution to macrostructure narrative processing when regions related to memory exhibit a decreased contribution with age. This finding suggested a compensatory interaction between these two cognitive processes. While language and space exist as distinct entities in our system of representation, they intricately interact. Even some researchers propose that language is spatial[72]. In social communication, speakers often anchor their utterances to their spatial

environment[73]. This anchoring is referred to as "deixes". Guided by deixes, narrators can effectively convey discourse context and enhance narrative skills, facilitating audience comprehension[73,74]. In information representation, theorists posit that various materials manifest diverse conceptual framework, with language and space acting as quintessential examples[75]. Both narrative and spatial representations span the spectrum from concrete to abstract. At the concrete level, the representation of space and language differs from each other. Spatial representations are typically perceptual, geometric, and sensorial, while language tends to be conceptual, algebraic, and amodal. Nevertheless, at the abstract level, both space and language need to extract schema from perceptual or event information, potentially leading to convergence in the abstract realm concerning conceptual structures and spatial patterns[75,76]. This is also the reason why, some individuals with spatial constructional deficits have shown normal scores in microlinguistic assessments but demonstrate weaknesses in macrostructure narrative capacity[77]. This finding suggested that there is some relationship between the cognitive mechanism of spatial ability and macrostructure narrative ability. This mutual effect is mainly reflected in the following two aspects.

Also, according to situation models, adults tend to update spatial location information that is consistent with the protagonist's perspective while processing narrative tasks[78,79]. In other words, this ability to grasp spatial information serves as a strategy to support the formation of a narrative framework. Corresponding to the compensatory effect observed in the present study, the aging brain adopts a strategy to increase the contribution of brain regions involved in spatial perception to the narrative process to delay the rapid decline in narrative ability. Moreover, updating spatial location information in narrative framework formation requires less intensive cognitive resources than updating other types of information, such as temporal series information[80]. Thus, the compensatory effect of spatial-related brain regions contributing to the narrative framework is the most significant and effective.

Furthermore, cortical spatial constructional processing is universally divided into ventral and dorsal pathways[81]. According to our results, the brain regions involved in the dorsal pathway, such as the posterior parietal lobe and the intraparietal sulcus, tended to be compensatory regions. The dorsal pathway was more likely to be labeled the pathway that supported both spatial perception and nonconscious spatial processing and guided individual behavior and actions[81]. This ability to organize cognitive units coincides with the macrostructure of narrative. Therefore, the dorsal pathway, especially the posterior parietal region, theoretically contributes to the central mechanism underlying macrostructure narrative ability. It is reasonable to consider these regions as compensating for the decline in narrative ability associated with aging.

Future studies can be conducted to address the following aspects. First, it would be valuable to investigate whether the compensatory effect associated with spatial constructional ability can be generalized to longitudinal data, allowing for the examination of changes over time. Second, because the participants in the present study were all healthy older adults, it remains unclear whether the compensatory mechanism is specific to non-pathological aging. It is important to ascertain whether individuals experiencing pathological aging are capable of compensating for the decline in narrative ability through similar neural substrates. Third, understanding the factors that influence the extent of this compensatory response and whether it effectively delays the decline of narrative ability is essential.

In summary, this study offers insights into the cognitive and neural factors underlying narrative abilities and proposes a potential compensatory mechanism involving brain regions associated with spatial constructional processing. This perspective offers a novel approach to comprehending the nature of narratives and their changes during the aging process.

## Methods
### Participants
Seven hundred and forty participants (533 females) aged 50 to 90 years (mean age 68.26 ± 8.20 years) participated in the first behavioral part of the study. The inclusion criteria were as follows: native Mandarin speaker, score ≥24 on the Chinese version of the Mini-Mental State Examination (MMSE), and the ability to complete a battery of neuropsychological tests. Individuals who had a history of major neurological or psychiatric illness were excluded.

The above participants also underwent fMRI scans. The participants had high-quality resting-state fMRI and T1 MRI data.

### Stimuli and procedures
**Narrative**. A four-panel comic was used in the study as the narrative material (Supplementary Fig. 1). The comic depicts a story about an old woman who fell down accidentally when she got off the bus and received help from the surrounding people. The instructions required the subjects to familiarize themselves with the pictures first; after the experimenter confirmed that the subjects were ready to tell the story, they were asked to start the recording. The experimenter also asked the participants to confirm the completion of the story and stop the recording. All the text materials were transcribed by an iFlytek machine and then manually proofread by a psychological undergraduate, who is a native Mandarin speaker. The undergraduate student was not aware of the status of the participants. The first author, also a native Mandarin speaker, coded the samples; at the time of coding, the first author was also blinded to the demographic information of the participants.

**Cognition**. As described in our previous study, all participants underwent a battery of neuropsychological tests at baseline[82]. The assessment involved general cognitive ability and cognitive function across five domains, namely, memory, language, attention, visuospatial abilities, and executive function. General cognitive ability was tested using the Chinese version of the Mini-Mental State Examination (MMSE)[83]; memory was tested using the Auditory Verbal Learning Test (AVLT)[84]; executive function was tested using the Trail Making Test (TMT)[85]; spatial constructional ability was tested using the ROCF-Copy test[86]; and language was tested using the Verbal Fluency Test (VFT)[87].

### Coding
**Macrostructure measures**. Deconstructing the overall organization of the story allowed us to create corresponding scoring standards for seven macrostructure elements, namely, character, setting, initiating, event, internal response, plan, action series, and consequence. Each element was scored on a scale of 0–3 points[88,89].

The detailed grading criteria are presented in the supplementary materials (Supplementary Table 1). According to the rules, the frequency of each macrostructure element described by the participants, as well as the connection of the element to the main storyline, was taken into account. The scoring of different elements focused on the subjects' grasp of the causality of materials, which can better reflect the narrative modes of different subjects.

**Microstructure measures**. The microstructures used in this study included the total number of characters or words (TNC/TNW), total number of different characters or words (NDC/NDW), longest sentence length (LSL), average sentence length (ASL), average character frequency (ACF) and average word frequency (AWF), as well as grammatical errors, fluency problems and content errors. TNC and TNW referred to the original number of words in the narrative recording transcribed into text; NDC and NDW referred to the number of characters and words that are not repeated in the text; and LSL was defined as the number of words used in the sentence with the most words in the text. The subjects described picture 1 and picture 2 as one sentence and picture 3 and picture 4 as another sentence; ASL was calculated by adding and averaging on this basis. ACF and AWF were defined as the sum of the frequency of each character or word used in the text. The richer and more diverse words used, the lower the average frequency was. Finally, grammatical errors, fluency problems and content errors were the statistics of the errors in the narrative text of the participants. If an error in a sentence appeared once, the number of grammatical errors was recorded as 1, with a maximum of

3 points. Similarly, fluency problems were defined as the number of pauses during the narrative, and content errors were defined as descriptions inconsistent with the picture contents. The above scores were generated automatically via Python.

## Image acquisition and data preprocessing

**MRI data acquisition.** MRI data, including T1-weighted MRI and resting-state functional magnetic resonance (fMRI) scans, were acquired via a Siemens Trio 3T scanner at the Imaging Center for Brain Research at Beijing Normal University. During the scans, participants lay supine with their head snugly fixed by straps and foam pads to minimize head movement and were instructed to stay awake and relax with their eyes closed. High-resolution T1-weighted, sagittal 3D magnetization-prepared rapid gradient echo sequences were acquired and covered the entire brain (176 sagittal slices, repetition time = 1900 ms, echo time = 3.44 ms, slice thickness = 1 mm, flip angle = 9°, inversion time = 900 ms, field of view = 256 mm × 256 mm, and acquisition matrix = 256 × 256). Resting-fMRI data were acquired using a gradient echo EPI sequence (TE = 30 ms, TR = 2000 ms, flip angle = 90°, 33 slices, slice thickness = 4 mm, in-plane matrix = 64 × 64, field of view = 256 × 256 mm$^2$). The resting scans lasted for approximately 8 min, and 240 image volumes were obtained.

**T1-weighted MRI preprocessing.** The SPM12 and CAT12 (http://www.neuro.uni-jena.de/cat) toolboxes were used to preprocess the anatomical images. Preprocessing was performed with the default settings, including high-dimensional diffeomorphic anatomical registration through exponentiated lie algebra (DARTEL) normalization algorithms and modulation of nonlinear components. Preprocessing steps also included the segmentation of whole-brain images into GM, WM, and cerebrospinal fluid and the normalization of the DARTEL template to the Montreal Neurological Institute space (Template_1_IXI555_MNI152.nii). Finally, we estimated the total intracranial volume (TIV) and the absolute volume of GM/WM using the "Estimate Total Intracranial Volume" module of CAT12. All GM and WM maps were smoothed by an 8 mm full width at half-maximum (FWHM) kernel.

**Resting-fMRI preprocessing.** For each participant, the first 10 volumes were discarded to allow for adaptation to the magnetic field. Resting data were preprocessed using Statistical Parametric Mapping (SPM; http://www.fil.ion.ucl.ac.uk/spm/), including slice timing, within-subject interscan realignment to correct possible movements, spatial normalization to a standard brain template in the Montreal Neurological Institute coordinate space, resampling to 3 × 3 × 3 mm$^3$, and smoothing with an 8 mm full-width at half-maximum Gaussian kernel. In addition, resting-fMRI data were processed with linear detrending and 0.01–0.08 Hz bandpass filtering and regression correction for nuisance covariates, which included six motion parameters, the global mean signal, the white matter signal, and the cerebrospinal fluid signal.

**Cortical parcellation and region of interest (ROI) definition.** To estimate functional connectivity from resting-state fMRI data, we first parcellated the brain into 400 parcels (i.e., nodes) according to the Schaefer 2018 parcellation atlas matched to Yeo 7 networks[90]. Behavioral data suggest a strong correlation between narrative ability and various cognitive processes, particularly episodic memory, executive function, and spatial constructional abilities. Additionally, prior research has demonstrated that the ventromedial prefrontal cortex, orbital frontal cortex, anterior cingulate cortex, and default mode network are the main regions involved in narrative ability. Therefore, we selected the default mode network (DMN), frontoparietal network (FPN), dorsal attention network (DAN) and ventral attention network (VAN) as regions of interest and excluded networks that do not represent high-level cognitive features of narrative ability, such as the visual network[91–94]. For each participant, the degree of connectivity was estimated by calculating the Pearson correlation coefficient of the 236 nodes' BOLD time series, which resulted in 27,730 edges.

**Data reliability.** The majority of microstructures were generated automatically by the same code. However, the macrostructure scores and the three kinds of errors needed to be validated. Following the same rubric, another undergraduate majoring in psychology independently scored 20% of the text materials that were randomly selected from all narratives. Cohen's k (a statistic measuring interrater agreement for qualitative data) indicated that agreement between the two coders was substantial ($k = 0.611$, $p < 0.0001$)[95].

## Statistical analysis

**Behavioral data.** The relationship between the two kinds of narrative indicators and cognitive functions was investigated by a multiple linear regression model. Additionally, with macrostructure and microstructure as dependent variables, a seemingly uncorrelated regression (SUR) model was used to test the significant difference between the coefficients of these two regression models to detect dissimilar patterns of cognitive interpretation[96]. Furthermore, we used the Sharpley value for the presentation of multidomain cognitive functions, which contributed to the explanatory power of the regression model[97].

In addition, regarding the trajectory of narrative aging, a generalized additive model (GAM) was used to estimate the nonlinear trend to find the inflection point at the group level. We used the SUR model to evaluate the differences in the coefficients between two subgroups divided according to age (older age and middle-aged patients).

## Neuroimaging data

**Linking macro- and micro-structures of narratives with functional connectivity patterns.** We described an algorithm inspired and modified from the CPM protocol[41] for selecting narrative correlated edges based on a set of single-subject connectivity matrices, constraining these edges using cross-validation of the testing set and finally constructing a predictive model to test the generalization of these edges (schematic shown in Fig. 1). Through this approach, the specific brain regions associated with narrative components were identified. Then, these contributing edges could be constructed as an undirected network. We ranked nodes according to the authority value of network topological measurements[98] and extracted the top 50 nodes as the primary brain regions representing the narrative components. Additionally, to evaluate the neural disassociation between different narrative structures, we calculated the degree of overlap of the primary nodes contributing to the narrative performance, which was quantified by Dice similarity coefficients[99]. In addition, we used conjunction analysis to visualize common or specific nodes that represented macro- and microstructure narrative components.

In addition, we performed community detection on the undirected weighted network using the Louvain algorithm[100,101]. Based on the principle of maximizing modularity, the Louvain algorithm calculates a greedy algorithm to minimize the number of edges within communities and maximize the number of edges between communities. During each iteration, the algorithm assigns nodes to communities, optimizing the modularity of each community and obtaining the optimal community allocation result. The community assignment results were subsequently subjected to hierarchical clustering via the linkage function, with Ward's method serving as the distance measure index. In the present study, four communities were identified.

**Validating the higher functional hierarchy of macrostructures mapped according to RSFC.** In the hub brain regions (nodes) representing the narrative components, we estimated the enrichment patterns of the contributing edges of the corresponding network connected to these nodes. The enrichment fold was calculated as the ratio of the actual observed number of selected edges within the network (Ai) to the expected number of selected edges (Ei). Ai was defined as the number of edges by which the node was

actually connected to the target network. Ei was calculated by multiplying the total number of edges of a node connected to all networks by the number of nodes in the target network and dividing by the maximum number of edges that can connect to that node (i.e., 236)[42,102]. We selected the top 10 hub nodes for the macro- and microstructures and calculated the enrichment fold using the contributing edges associated with them. We also independently calculated the enrichment fold using the contributing edges of representative nodes located in different brain regions.

Similarly, the enrichment fold of the community was calculated as described for the hub nodes, the actual observed nodes within the specific network divided by the expected number of nodes. Here, Ai refers to the number of nodes in a specific network actually included in the target community, and Ei refers to the total number of nodes in the target community multiplied by the actual number of nodes in the specific network divided by the total number of nodes.

The permutation test for calculating enrichment significance involved the randomization of the labels of nodes within a community and their assignment to different communities while maintaining the size of the original community. For example, if the main community comprises 90 nodes, it would still contain the same number of nodes.

We used the BrainSpace toolbox to identify, visualize, and analyze the large-scale gradients of brain organization among different narrative performance groups[103]. In the first stage, we identified the top third of participants according to their macro- and microstructure scores while excluding those who overlapped. Next, the RSFC patterns of these two groups were used as the input matrix to generate gradients. High-dimensional RSFC data were mapped onto a low-dimensional manifold using the default PCA linear dimensionality reduction method, and the gradients of all participants served as templates for gradient alignment via Procrustes analysis. Finally, we conducted independent sample T tests on the gradient values of each node of the two aligned groups of participants and identified the brain regions with significant differences.

Investigating changes in the RSFC patterns associated with narrative aging using sliding age windows. The analysis used the hub brain regions representing the narrative components as seeds to construct the functional connectivity network of each subject. First, participants were arranged in ascending order according to their age and divided into consecutive age windows of 30 people each. Second, we constructed the RSFC matrix of each participant using the macro- and micro-structure hub nodes as seeds. Third, we computed the correlation matrix between the RSFC and narrative scores among participants in each age window. Finally, we related the RSFC-narrative correlation matrix to the mean value of the age window and used a permutation test to test for significance (false discovery rate [FDR] BH corrected, $P < 0.05$), which yielded the trend of the contribution of the related edges to narrative changes with age.

We did not use a fixed age range as a time window to avoid unnecessary errors caused by a significant difference in the number of participants in different age windows (because the age distribution of the subjects was close to normal). However, we verified that there was no difference between these two methods in exploring the neural correlate of narrative aging (Supplementary Fig. 2).

Validating three methods for investigating cognitive function represented by narrative aging-related brain regions. Fit linear regression model: Based on the RSFC pattern of narrative aging, a functional connectivity matrix was constructed between each subject's narrative hub nodes and specific brain regions (such as positive regions of macrostructures). Then, cross-validation was performed by randomly selecting 74 subjects as the test dataset and the remaining subjects as the training dataset. Next, linear regression model was established with the functional connectivity matrix as the independent variable and various cognitive functions as the dependent variables. In addition, the model used cross-validation to evaluate the performance. The model was fitted using the training dataset and used to predict the test dataset. Finally, the correlation coefficient between the predicted cognitive scores and actual cognitive scores was calculated as the evaluation index for the model.

Neurosynth decoding: We decoded the specific regions using meta-analyses in Neurosynth[104] (https://www.neurosynth.org/) and selected the first 30 related terms. Only one entry with the same meaning was retained (e.g., inferior frontal and inferior, keeping the former), and a word cloud of the remaining entries was generated based on their correlation value.

CPM analysis: The top 50 hub nodes for each cognitive function were identified using the flowchart in Supplementary Fig. 5. The degree of overlap between these nodes and specific brain regions was calculated using the Dice similarity coefficient. A permutation test was used to calculate the significance of similarity in the Dice coefficient.

Using multimodal structural data to validate the RSFC pattern in narrative aging. Furthermore, we aimed to provide structural brain validation for the functional neural associations in narrative aging. We extracted the gray matter volume in each node and constructed the structural covariant network of all the subjects (with the hub cluster as the seed point and the nodes associated with narrative aging as the mask). The structural covariant network was correlated with the mean age in each window (successive age windows as above), and the development trend of each edge with increasing age and the structural connectivity of the seed points was obtained. The significant edges after correction were included in the subsequent analysis.

In addition, we selected fiber number as an indicator to construct a white matter network. The edges of the white matter network linked the narrative hub and the nodes involved in narrative aging as the independent variation, while the correlation between these edges in the resting-state functional connectivity network and age was used as the dependent variation. We constructed a high-dimensional regression model to predict functional connectivity patterns associated with narrative aging using white matter structures. First, we normalized the data and divided it into training and testing sets. Second, the independent variation and dependent variation were calculated in both the training and test sets. We used the variations in the training set to construct a linear regression model using the cross-validation method to tune the hyperparameters. Finally, we used the correlation between the predicted and actual results to evaluate the model performance. The predicted response variable was calculated on the basis of the test set data by using the regression coefficient and intercepts of the linear model.

## Reporting summary
Further information on research design is available in the Nature Portfolio Reporting Summary linked to this article.

## Data availability
The Neurosynth database is available at https://neurosynth.org/. Raw data of the older adults with their completed narrative and extensive neuropsychological tests and MRI scans are available from the corresponding author on reasonable request. Restriction of raw data is to protect the privacy of participants. Source data are provided with this paper.

## Code availability
Custom codes are variable at https://github.com/Rainmon2020/Narrative-Aging2022.

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

## Author contributions

L.Y.M. and L.X. conceived the study. L.Y.M. participated in the data collection. L.Y.M. carried out statistical analysis. L.Y.M. analyzed and interpreted the data. L.Y.M. wrote the manuscript. L.X. revised the manuscript. Z.Z.J. and Z.J.Y. supervised and coordinated the study. All authors contributed to the improvement of this manuscript and approved the final version for submission.

## Competing interests

The authors declare no competing interests.
