## [Transparent Peer Review file · Communications Biology]

Uncovering narrative aging: An underlying neural mechanism compensated through spatial constructional ability

Corresponding Author: Professor Xin Li

Version 0:

Reviewer comments:

Reviewer #1

(Remarks to the Author)

Thank you for the opportunity to review this interesting manuscript. Narrative aging is an understudied topic used to investigate the experience of aging that incorporates wider social and cultural values. The current study investigated the expression of aging through this narrative process by focusing on high-order functions (macrostructure) and language specificity as a microstructure. To achieve this aim, the authors used a story task with verbal and visual stimuli to investigate the neural correlates of these macro-and-micro structure aspects of narrative aging in an fMRI study with a sample of $n = 740$ older participants. A lot of results are reported in the paper, but in summary, key findings included that macrostructures identified by the task were significantly more related to various cognitive measures. For example, there was an increase in spatial construction ability but a decrease in episodic memory, and in general macro structures showed a general decline and were more vulnerable to aging. In comparison, the microstructures of narrative aging remained relatively stable in aging and were specifically associated to language function. The neural correlated found are a little unclear to me, but significant "hubs" of interaction in the angular gyrus and the medial frontal lobes for the macrostructure, the supramarginal gyrus and the middle cingulate cortex for the microstructure.

Overall, this study has elements of novelty and has potential. However, I found the paper difficult to read and follow, with the exact aims being difficult to uncover and with a plethora of results that did not always match the exact aims of the study. I have added some more specific suggestions below to help add a little more clarity and specificity to the paper:

1. The abstract at present is a little unclear; it could be restructured to match the aims presented in the introduction.
2. For the methods, more details in the demographic section of the large sample is needed, other important factors like, socio-economic status, level of education and age itself could be added as covariates in the analysis.
3. More detail is needed on the neuropsychological tests used together with the normative data, maybe summarised in a table etc.
4. There were so many results reported, it was difficult to identify the key findings of the paper. I would suggest that the results section also be restructures, as to identify only the key findings that are in line with the aims and hypotheses that should be specifically stated in the introduction. The remaining results can be added in the supplementary materials perhaps.
5. The results around the possible shared neural correlates of macro and micro structures are very interesting, but again, so many results were reported, it is easily to lose track of the main results. I wonder if the analysis could be simplified, as restructured to identify the core findings and also eliminate the possibility of over testing. In a study such as this with so many variables, I think it is important to make sure the aims and hypothesis are very clear and the analysis conducted match those specific aims, rather than analysing a whole range of variables.
6. The behavioural results for example are in itself very interesting in finding differences in cognitive ability between macro-and-micro structure in narrative aging. The suggestion that spatial construction ability could act as a compensator mechanism to ageing and dementia is a key finding that requires more detail in the results section. Whereas, the fMRI results are very unclear, maybe because of the extract of the results reported.
7. Some of the background information provided in the discussion, might be more useful to add in the introduction.

In summary, the focus on narrative aging is a novel component of the study and it shows great promise, however, as it

stands, the aims and results of the study are not entirely clear and this undermines the impact of the results reported.

Reviewer #2

(Remarks to the Author)

Overall, this study is strengthened by an interesting design with relatively substantial sample size. The findings provide novel evidence that the cognitive mechanism and neural associations that represent narrative aging using macro- and microstructure. The manuscript is easy to read and understand. Nonetheless, I found several concerns and issues listed below which need to be addressed and clarified by the authors.

Abstract

Add the demographic characteristics of the participants in abstract.

Introduction

Line 76 "The narrative itself involves multiple cognitive operations which exhibit distinct patterns of aging." According to the theory, what are the characteristics of narrative ability during the lifespan? Are there the differences of cognitive functions related to narrative ability between older adults and children?

Line 110 "narrative is relatively complex and coherent and contains rich meaning, the higher-order cortical networks, such as DMN and PMN" Narrative represented the language production to generate a framework of meaning. However, why the cortical networks do not include the typical region which involved in language?

Add the details about the definition of macrostructures and microstructures of narrative.

Results

Line 321 In section of Dice similarity coefficients, the range of Dice coefficient is 0.15-0.31. These coefficients are not very high. The results of the identification the hub nodes associated with different cognitive functions are not very convincing.

Discussion

The discussion section stretched out over six pages, however, to my opinion, the related explanations for neural substrates and of macro and micro structure occupied much. I suggested the authors to cut out some unnecessary expressions.

Materials and methods

Line 567 The author's approach to quantifying narrative ability as overly simplistic. Nowadays, numerous natural language algorithms can be employed to assess textual content, highlighting that the author's methodology may not offer significant advantages. The author should elucidate why this quantification method was chosen.

Line 699 In section "Linking macro- and micro-structures of narratives with functional connectivity patterns", "... these contributing edges could be constructed as an undirected network..."it would be more informative to report the functional connectivity network patterns

Line 764 "Participants were arranged in ascending order according to their age and divided into consecutive age windows of 30 people each." The author explores to the relationship between narratives and age, however, why divided the age windows based the sample of participants instead of age (such as 5-years or 10-years)?

It would be beneficial for clarity and transparency to include specific details about the Ethics Committee that approved the study. Providing the name or affiliation of the committee would enhance the credibility and accountability of the research process. Could you please consider revising the sentence to include more specific information, such as the name of the Ethics Committee or the institute to which it belongs?

I would like to suggest some minor improvements to enhance the overall clarity of the document. Specifically, I recommend that you thoroughly review the entire draft, including the abstract, for typos and grammatical errors.

Version 1:

Reviewer comments:

Reviewer #1

(Remarks to the Author)

Thank you to the authors for the substantial changes made to the paper that have greatly enhanced the quality of the manuscript. The cover letter is very clear and the changes explained in the letter are very useful. However, I apologies if I missed anything, but I think the changes made in the actual manuscript have not been highlighted. These changes have

been noted in the cover letter, which is very useful, but I think more have been made in the actual paper. I would like to see a version of the revised manuscript that includes the tracked or highlighted changes in the revised paper. Otherwise, I have a few more comments on this revised version:

1. I would suggested added inverted commas -i.e., "The narrative" in the start of the abstract and in the introduction, to make the actual topic of study clear, otherwise, it sometimes looks like a language mistake inadvertently, and it might be hard for authors to realise that "the narrative" and narrative processes in ageing, (i.e. what is called narrative ageing in the paper) is what is being investigated here.
2. From the abstract now, it seems like a pre-existing dataset is being used for this study, but in the methods section it sounds like first hand data-collection has been conducted, this needs to be clarified throughout the paper.
3. As the other reviewer has noted, the sample size is a huge strength of the study, and this can be highlighted throughout the paper (e.g., abstract, introduction, results etc.).
4. The three overarching aims of the study are very useful to have and thank you to the authors for including this. The results, I think, have also now been restructured in this way, and the use of subheadings helps guide the reader (as the results are still very extensive). It might be useful to have each of these research questions either in an additional overarching subheading in the results, or in the narrative write up of the results, explaining what research aim or question the specific set of results is referring to.

Minor changes:

1. in the abstract, MRI scan needs to be plural - MRI scans or just say MRI methods.
2. all p-values need to be in small letters and in italics, or the authors need to be consistent in the way they report this.
3. Effect sizes need to be consistently reported please.

Reviewer #2

(Remarks to the Author)

The authors have adequately addressed my concerns, and I have no further questions.

Responses to Reviews

(Submission ID: COMMSBIO-24-2253)

Reviewer #1 (Remarks to the Author):

Thank you for the opportunity to review this interesting manuscript. Narrative aging is an understudied topic used to investigate the experience of aging that incorporates wider social and cultural values. The current study investigated the expression of aging through this narrative process by focusing on high-order functions (macrostructure) and language specificity as a microstructure. To achieve this aim, the authors used a story task with verbal and visual stimuli to investigate the neural correlates of these macro-and-micro structure aspects of narrative aging in an fMRI study with a sample of $n = 740$ older participants. A lot of results are reported in the paper, but in summary, key findings included that macrostructures identified by the task were significantly more related to various cognitive measures. For example, there was an increase in spatial construction ability but a decrease in episodic memory, and in general macros structures showed a general decline and were more vulnerable to aging. In comparison, the microstructures of narrative aging remained relatively stable in aging and were specifically associated to language function. The neural correlated found are a little unclear to me, but significant “hubs” of interaction in the angular gyrus and the medial frontal lobes for the macrostructure, the supramarginal gyrus and the middle cingulate cortex for the microstructure.

Overall, this study has elements of novelty and has potential. However, I found the paper difficult to read and follow, with the exact aims being difficult to uncover and with a plethora of results that did not always match the exact aims of the study. I have added some more specific suggestions below to help add a little more clarity and specificity to the paper:

1. The abstract at present is a little unclear; it could be restructured to match the aims presented in the introduction.

Response: Thank you for your helpful suggestions. We have revised the abstract section to highlight the purpose of this paper. About what you mentioned: “it could be restructured to match the aims presented in the introduction.” In the revised abstract section, we will provide explanations of the results concerning the three aims proposed in the introduction.

Aim1: We will investigate whether the macrostructure involves more complex nonlinguistic cognitive processes than microstructure. This is evidenced by the fact that the resting-state functional connectivity pattern of macrostructure is located at a higher functional gradient level. This aim corresponds to the line 32-39 in abstract.

Aim2: The macrostructure is more susceptible to the effects of aging. Thus, we want to investigate the potential cognitive contribution and underlying neural correlates of macrostructure narrative aging. This aim corresponds to the line 39-44 in abstract.

Aim3: The neural substrates supporting narrative aging have a corresponding structural basis. This aim corresponds to the line 44-46 in abstract.

2. For the methods, more details in the demographic section of the large sample is needed, other important factors like, socio-economic status, level of education and age itself could be added as covariates in the analysis.

Response: Thank you for your constructive suggestions. The following table provides additional information on sociodemographic variables and neuropsychological test performance for all participants.

	N	Mean	SD	Min	Max
Age	740	68.26	8.20	50	90
Education	740	12.14	3.04	6	19
Episodic Memory	740	25.81	9.19	0	55
Spatial-Construction	740	32.6	5.01	0	36
Processing speed(time/s)	740	70.9	27.88	23	258
Executive function(time/s)	740	182.58	79.29	52	566
Verbal fluency	740	44.64	10.35	13	90
General cognition	740	28.28	1.51	24	30

Note: In this sample of 740 individuals, there were 533 females and 207 males. The units for age and education level in the table are years, while the units for performance in different domains of neuropsychology are scores. Only the units for processing speed and executive function are in time/seconds, representing the time taken to complete the tests.

3. More detail is needed on the neuropsychological tests used together with the normative data, maybe summarised in a table etc.

Response: Thank you for your suggestions. We completely agree with your opinion. We have added the scores of neuropsychological tests in the figure above, and we will also present this information as Supplementary Table 2.

4. There were so many results reported, it was difficult to identify the key findings of the paper. I would suggest that the results section also be restructures, as to identify only the key findings that are in line with the aims and hypotheses that should be specifically stated in the introduction. The remaining results can be added in the supplementary materials perhaps.

Response: Thank you very much for your comments. We are sorry for the confusion caused by our expressions in the Result section. We have adopted your suggestions, reorganized our results, and included some confirmatory results in the supplementary materials. We hope that the revised manuscript can more clearly articulate our research objectives and hypotheses.

Our specific editing steps are as follows:

1. Firstly, we moved some confirmatory results to the supplementary materials, mainly the original results "Descriptive statistics and bivariate correlations between items," which describe the measurement effectiveness of each item in macrostructure and microstructure. We also relocated the use of group gradient indices in the original result 2.5.3 to further confirm the results of macrostructure being at a high functional level.
2. Secondly, we also relocated expressions that verify the same result using different methods to the supplementary materials, such as in result 2.1, where Sharpley analysis, similar to SUR analysis, explains the contribution of different cognitive functions to variations in narrative structure degrees.

3. Moreover, to present the main framework of the results clearly, we split the previously complexly expressed result section 2.6 into two clearer parts: result section 2.5 aims to explain the possible neuro-compensatory effects in narrative aging, while result section 2.6 decodes the corresponding brain regions of compensatory effects for spatial construction abilities.

4. Lastly, we moved figure 2, which was not significantly related to the main results framework, to the supplementary materials. This figure is solely used to illustrate the specific process of the CPM method operation.

The aforementioned are the significant changes made regarding the results framework in this paper. Additionally, we have revised some detailed expressions in the manuscript to enhance clarity, which are not elaborated on here. We hope that the restructured manuscript will highlight the key results of this study more effectively.

5. The results around the possible shared neural correlates of macro and micro structures are very interesting, but again, so many results were reported, it is easily to lose track of the main results. I wonder if the analysis could be simplified, as restructured to identify the core findings and also eliminate the possibility of over testing. In a study such as this with so many variables, I think it is important to make sure the aims and hypothesis are very clear and the analysis conducted match those specific aims, rather and analysing a whole range of variables.

Response: We appreciate your insightful comment on the overall presentation of our results in this study. The analysis in the main manuscript has been simplified and rearranged as per your suggestion 4. However, we need to explain that because we included a large sample of elderly participants and the elderly population exhibits significant individual differences, our analysis methods involve cross-validation, and the presentation of key results also includes corresponding confirmatory analyses. There may have been some confusing parts in how we presented results and organized the text, and we have made significant revisions.

Please allow us to briefly explain the main results of the fMRI section based on the hypotheses we have stated in the introduction:

1. Firstly, we need to verify hypothesis one, which states that "macrostructure involves more complex cognitive processes compared to microstructure." Through the CPM method, we investigated the neural substrates that separate macrostructure and microstructure supporting narratives (Figure 2) and calculated the brain networks enriched by these neural substrates (Figure 3). This confirms that both the neural substrates characterizing macrostructure and the brain networks they enrich are situated at higher functional hierarchy, demonstrating that macrostructure involves more complex cognitive processes compared to microstructure.

2. Secondly, to validate hypothesis two, which suggests that "due to its more complex cognitive processes, the mechanisms underlying aging effects on macrostructure differ significantly from microstructure," we found (Result 2.5) that certain brain regions play an increased role in the aging process of macrostructure (Figure 4). To decode the specific cognitive function associated with these brain regions (Result 2.6), we utilized three methods depicted in Figure 5 for decoding to achieve cross-validation. Ultimately, we discovered that these brain regions correspond to spatial construction abilities.

3. Finally, in order to verify hypothesis three, which indicates that "the neural correlates supporting the aging of narrative abilities have a certain structural basis," and further confirm that spatial construction abilities can compensate for aging effects on macrostructure, a phenomenon not

explained by microstructure. The results (Result 2.7) showed that only the structural connectivity of the neural substrates characterizing macrostructure and compensatory regions (decoded as brain regions representing spatial construction abilities) is stronger.

6. The behavioural results for example are in itself very interesting in finding differences in cognitive ability between macro-and-micro structure in narrative aging. The suggestion that spatial construction ability could act as a compensator mechanism to ageing and dementia is a key finding that requires more detail in the results section. Whereas, the fMRI results are very unclear, maybe because of the extract of the results reported.

Response: Thank you very much for your thorough reading of our manuscript and for acknowledging our results. We are sorry for any confusion caused by the expression in the fMRI section. We also find it very interesting that the spatial construction ability reflected in our behavioral results may serve as a compensatory mechanism for narrative aging. Therefore, the logical flow of presenting our fMRI results is not only to extend our research hypothesis but also to provide evidence at the neural connectivity level for the spatial construction ability as a compensatory mechanism for narrative aging. Our results in sections 2.5 and 2.6 specifically elaborate on these arguments.

1. Result 2.5: We found that with increasing age, the warm-colored brain regions in the Figure 4 contribute more to the representation of the macrostructure of narratives, which we refer to as the "compensatory brain regions". The compensatory brain regions were recruited because their RSFC representing the macrostructure increased with aging.
2. Result 2.6: In order to decode the function corresponding to the "compensatory regions," we utilized three decoding methods from Figure 6. Through cross-validation of the three methods, it was revealed that the representation of the macro-structure of the compensatory brain region signifies spatial construction ability.
3. Result 2.7: In order to validate that the compensatory effect only exists in the macro-structure (as compensatory regions also exist in the micro-structure in the figure above), we verified the brain structures of the participants, including gray matter and white matter. We calculated the structural connectivity of hub areas and compensatory regions in different narrative structures, including gray matter co-variation and white matter connections. The results indicate that only the compensatory effect in the macro-structure has a structural basis.

7. Some of the background information provided in the discussion, might be more useful to add in the introduction.

Response: Thank you for your constructive suggestions. We carefully examined the discussion section, and we understand that your suggestion is to remind us that the theories related to spatial ability and language ability mentioned in lines 466-476 are suitable to be placed in the introduction to provide a leading guide to the results. We appreciate your thoughtful suggestion and will provide an explanation for our decision to include this section in the discussion. The purpose of our reference to the situation model is to explain a relatively novel finding we discovered: the potential compensatory effect of spatial construction ability in narrative aging. Unfortunately, there were not sufficient prior assumptions in the introduction for this result. Its initial hypothesis was based on our behavioral results because the potential interaction between spatial ability and language ability was a relatively innovative viewpoint for us. Therefore, we subsequently designed many statistical

frameworks including cross-validation to explain this result from the perspective of brain substrates. Fortunately, this result passed multiple cross-validations, demonstrating robustness. As the compensatory effect of spatial ability was not the research question we posed when designing the experiment, we did not include this theory in the introduction. We hope you understand.

In summary, the focus on narrative aging is a novel component of the study and it shows great promise, however, as it stands, the aims and results of the study are not entirely clear and this undermines the impact of the results reported.

Reviewer #2 (Remarks to the Author):

Overall, this study is strengthened by an interesting design with relatively substantial sample size. The findings provide novel evidence that the cognitive mechanism and neural associations that represent narrative aging using macro- and microstructure. The manuscript is easy to read and understand. Nonetheless, I found several concerns and issues listed below which need to be addressed and clarified by the authors.

Abstract

1. Add the demographic characteristics of the participants in abstract.

Response: Thank you for your constructive suggestions. We have added the demographic information in the abstract section (page 1, line 31-32). We have also added the detailed demographic information in the supplementary materials.

Introduction

2. Line 76 “The narrative itself involves multiple cognitive operations which exhibit distinct patterns of aging.” According to the theory, what are the characteristics of narrative ability during the lifespan? Are there the differences of cognitive functions related to narrative ability between older adults and children?

Response: Regarding the narrative ability mentioned in this article, there is currently no systematic result on the age-related changes in narrative ability features within the lifespan range. Therefore, it is not possible to interpret it from a relatively macro perspective. I can only provide an analysis of the developmental trends of narrative ability at different stages. During early childhood (under 10 years old), the macro and micro features of narrative ability improve with age. Due to limited related research and participant numbers, no significant turning points have been found.

However, narrative ability is crucial in early development, and in school-age children, it greatly influences their non-verbal intelligence and academic achievements¹. Moreover, children with pathological early developmental characteristics such as speech disorders, learning disabilities, and autism show significantly lower narrative ability scores than typically developing children²⁻⁴. For adults, narrative ability is not frequently used in a series of assessments, such as tests for verbal intelligence and adult reading abilities. Furthermore, studies have shown that narrative therapy can significantly improve the quality of life for adult individuals with aphasia caused by traumatic brain injury compared to conventional interventions⁵. On the other hand, studies have shown that children

with language impairments do not show significant differences in macro-structure compared to typically developing children, but their micro-structure are noticeably inferior ⁶. These findings highlight notable variations in narrative ability structures across diverse groups, signaling vast opportunities for future research in the field.

3. Line 110 “narrative is relatively complex and coherent and contains rich meaning, the higher-order cortical networks, such as DMN and PMN” Narrative represented the language production to generate a framework of meaning. However, why the cortical networks do not include the typical region which involved in language?

Response: Thank you for your comments. This sentence aims to convey that the cognitive process of narrative is intricately complex, engaging sophisticated cortical networks and favoring macro-structure, hence employed here for evidential demonstration. In task-fMRI experiments with narrative as the main task, the DMN and PMN networks often appear as ROIs, serving as prior knowledge in this field ⁷⁻¹⁰.

Furthermore, previous studies have shown that in the narrative process, the DMN and PMN networks are used to represent events with relatively long-time scales (macroscopic), while other cortical regions biased towards perception represent units with shorter time scales (microscopic, such as syllables and words) ¹⁰. In addition, regarding the language-related network, recent research indicates that language production relies on the coordination of a large-scale brain network. Therefore, there is no specific brain region that can independently represent language functions in a general sense ^{11,12}. The prefrontal cortex and the related cortical regions in the temporal-parietal junction (important areas in the DMN) are also crucial in language-related brain networks. Thus, although the mention of DMN and PMN networks may be confusing, multiple pieces of evidence from studies suggest that this approach is rational.

4. Add the details about the definition of macrostructures and microstructures of narrative.

Response: The concept of macrostructure and microstructure of narrative is derived from interactive-construction model theory (page 3, line 71-78). *“According to the theory, the narrative has two structures. On the one hand, the formation of basic language structures requires interconnections between linguistic elements, which can be called microstructures.”* This sentence provides the theoretical definition of microstructure, exemplified by extracting features such as words and sentences, the relatively minute components, from the textual output *“On the other hand, the macrostructure emerges as an organizational representation of conceptual units, involving a top-down framework determining the logic of the entire narrative and reflecting the mind of the narrator”*. The definition of macrostructure involves extracting the top-down conceptual framework from individual narrative materials, reflecting the individual’s macro-logical perspective. The diagram below illustrates our framework for defining the micro and macro aspects of narrative structure, allowing for a deeper comprehension of narrative macro and micro structures through the details provided.

Interactive-construction Model

Results

5. Line 321 In section of Dice similarity coefficients, the range of Dice coefficient is 0.15-0.31. These coefficients are not very high. The results of the identification the hub nodes associated with different cognitive functions are not very convincing.

Response: I am sorry that the magnitude of the values has caused confusion. We fully understand your concerns. However, we also need to provide some explanations. We do not evaluate the results based solely on the magnitude of the Dice coefficient. Instead, we conduct a significance test for each Dice coefficient using the permutation method. Only Dice coefficients with significance greater than 0.05 after FDR correction are taken into consideration. We hope that by conducting significance tests, the Dice coefficients can enhance credibility for you.

Discussion

6. The discussion section stretched out over six pages, however, to my opinion, the related explanations for neural substrates and of macro and micro structure occupied much. I suggested the authors to cut out some unnecessary expressions.

Response: Thank you for your suggestions. We will carefully review the discussion section and remove some unnecessary discussions about macro and microstructural neural substrates.

Materials and methods

7. Line 567 The author's approach to quantifying narrative ability as overly simplistic. Nowadays, numerous natural language algorithms can be employed to assess textual content, highlighting that the author's methodology may not offer significant advantages. The author should elucidate why this quantification method was chosen.

Response: We highly agree with your opinions. Currently, with the rapid development of Natural Language Processing (NLP) technology, we can extract more narrative features based on various NLP algorithms for text processing. We have had discussions with Chinese companies specializing in intelligent speech and voice technology research, such as iFlytek. They are indeed able to automatically generate text features using models, although these features can be categorized as microstructure in this article. At present, there is no relatively perfect language model for outputting macrostructure, or this area has not received sufficient attention. Through this manuscript, we also aim to inspire and provide insights to more algorithm researchers. We also intend to integrate macro-level features with natural language algorithms, striving to design and extract text features that better align with the narrative macro structure in theory, and utilize them for the quantification in the aging development stage as well as early screening of normal aging and pathological aging. This is our

next research plan.

8. Line 699 In section “Linking macro- and micro-structures of narratives with functional connectivity patterns”, “.... these contributing edges could be constructed as an undirected network...”it would be more informative to report the functional connectivity network patterns

Response: Thank you for your suggestions. If the functional connectivity network patterns you mentioned refer to different types of brain networks, we have made it clear in the "Cortical parcellation and region of interest (ROI) definition" section (lines 651-652). Here, the functional connectivity network patterns actually refer to the separate neural associations reflected in macro and micro structure, so specific brain networks are not explicitly explained.

9. Line 764 “Participants were arranged in ascending order according to their age and divided into consecutive age windows of 30 people each.” The author explores to the relationship between narratives and age, however, why divided the age windows based the sample of participants instead of age (such as 5-years or 10-years)?

Response: Thank you for your helpful suggestions. Using a 10-year age window and an equal number of subjects window, we constructed the neural correlations of narrative aging, resulting in a high Dice similarity of 0.81. When changing the age window to a range of 5-10 years and the number of subjects in each window to 50-100, the outcomes remained consistent. The range of Dice coefficient values was 0.78 ± 0.23 , with a 95% confidence interval of [0.6339, 0.9261].

10. It would be beneficial for clarity and transparency to include specific details about the Ethics Committee that approved the study. Providing the name or affiliation of the committee would enhance the credibility and accountability of the research process. Could you please consider revising the sentence to include more specific information, such as the name of the Ethics Committee or the institute to which it belongs?

Response: Thank you for your reminding. The statement of approval by an ethical committee is the following: The study was conducted in accordance with the institutional review board (IRB) at the Imaging Center for Brain Research at Beijing Normal University (protocol code ICBIR_A_0041_002_02 and date of approval 03.2015).

11. I would like to suggest some minor improvements to enhance the overall clarity of the document. Specifically, I recommend that you thoroughly review the entire draft, including the abstract, for typos and grammatical errors.

Response: Thank you for pointing out our unsatisfactory language expression. We have thoroughly revised the expression throughout the entire paper and restructured the framework of the results to make the main findings clearer.

References

- 1 Shaqiri, A., Gallopeni, F. & Selmani, E. Non-verbal intelligence and the importance of group functioning in developing narrative skills and school success in children aged 7–9 years. *Early Child Development and Care* **192**, 775-780, doi:10.1080/03004430.2020.1800663 (2020).
- 2 Ferretti, F. *et al.* Time and Narrative: An Investigation of Storytelling Abilities in Children With Autism Spectrum Disorder. *Front Psychol* **9**, 944, doi:10.3389/fpsyg.2018.00944 (2018).
- 3 Reilly, J., Losh, M., Bellugi, U. & Wulfeck, B. "Frog, where are you?" Narratives in children with specific language impairment, early focal brain injury, and Williams syndrome. *Brain Lang* **88**, 229-247, doi:10.1016/S0093-934X(03)00101-9 (2004).
- 4 Tsimpli, I. M., Peristeri, E. & Andreou, M. Narrative production in monolingual and bilingual children with specific language impairment. *Applied Psycholinguistics* **37**, 195-216 (2016).
- 5 Corsten, S., Schimpf, E. J., Konradi, J., Keilmann, A. & Hardering, F. The participants' perspective: how biographic-narrative intervention influences identity negotiation and quality of life in aphasia. *Int J Lang Commun Disord* **50**, 788-800, doi:10.1111/1460-6984.12173 (2015).
- 6 Hao, Y. *et al.* A Narrative Evaluation of Mandarin-Speaking Children With Language Impairment. *J Speech Lang Hear Res* **61**, 345-359, doi:10.1044/2017_JSLHR-L-16-0367 (2018).
- 7 Lee, H., Bellana, B. & Chen, J. What can narratives tell us about the neural bases of human memory? *Current Opinion in Behavioral Sciences* **32**, 111-119, doi:10.1016/j.cobeha.2020.02.007 (2020).
- 8 Lee, C. S., Aly, M. & Baldassano, C. Anticipation of temporally structured events in the brain. *Elife* **10**, doi:10.7554/eLife.64972 (2021).
- 9 Chen, J. *et al.* Shared memories reveal shared structure in neural activity across individuals. *Nat Neurosci* **20**, 115-125, doi:10.1038/nn.4450 (2017).
- 10 Baldassano, C. *et al.* Discovering Event Structure in Continuous Narrative Perception and Memory. *Neuron* **95**, 709-721 e705, doi:10.1016/j.neuron.2017.06.041 (2017).
- 11 Yuan, Q. *et al.* Patterns and networks of language control in bilingual language production. *Brain Struct Funct* **226**, 963-977, doi:10.1007/s00429-021-02218-7 (2021).
- 12 Pyllkkänen, L. The neural basis of combinatory syntax and semantics. *Science* **366**, 62-66 (2019).

Responses to Reviews

(Submission ID: COMMSBIO-24-2253A)

Reviewer #1 (Remarks to the Author):

Thank you to the authors for the substantial changes made to the paper that have greatly enhanced the quality of the manuscript. The cover letter is very clear and the changes explained in the letter are very useful. However, I apologies if I missed anything, but I think the changes made in the actual manuscript have not been highlighted. These changes have been noted in the cover letter, which is very useful, but I think more have been made in the actual paper. I would like to see a version of the revised manuscript that includes the tracked or highlighted changes in the revised paper. Otherwise, I have a few more comments on this revised version:

Response: Thank you very much for your many suggestions that greatly improved the readability of our manuscript. We apologize for not submitting a version with track changes, and we will provide a marked version in this submission. Thank you for your reminding.

1. **I would suggested added inverted commas -i.e., "The narrative" in the start of the abstract and in the introduction, to make the actual topic of study clear, otherwise, it sometimes looks like a language mistake inadvertently, and it might be hard for authors to realise that "the narrative" and narrative processes in ageing, (i.e. what is called narrative ageing in the paper) is what is being investigated here.**

Response: Thank you for your helpful suggestions. We have revised all the expressions of “the narrative”. We have added quotation marks in places that could be easily misunderstood as per your request. (see line 54, 59 ,64, 77, 80, 109, 117)

2. **From the abstract now, it seems like a pre-existing dataset is being used for this study, but in the methods section it sounds like first hand data-collection has been conducted, this needs to be clarified throughout the paper.**

Response: Thank you for pointing out our less rigorous expressions. In the introduction, we simply aimed to highlight that the data collected in our study involved a large sample size and multidimensional behavioral-brain metrics. We apologize for any misunderstandings caused. We have revised the statements that may lead to confusion. (line 32)

3. **As the other reviewer has noted, the sample size is a huge strength of the study, and this can be highlighted throughout the paper (e.g., abstract, introduction, results etc.).**

Response: Thank you for the reminder. Relevant statements have been incorporated into the manuscript. (line 32, 387)

4. **The three overarching aims of the study are very useful to have and thank you to the authors for including this. The results, I think, have also now been restructured in this way, and the use of subheadings helps guide the reader (as the results are still very extensive). it might be useful to have each of these research questions either in an additional overarching subheading in the results, or in the narrative write up of the results, explaining what research aim or question the specific set of results is referring too.**

Response: Thank you for your constructive suggestions. In each section's summary of results,

we have added conclusive statements that validate the hypotheses. (line 209-212, 281-286, 366-373, 380) However, we did not make specific changes to the subheadings as they serve the methods used and conclusions drawn in this section, so unfortunately, we cannot provide further information.

Minor changes:

1. in the abstract, MRI scan needs to be plural - MRI scans or just say MTI methods.

Response: Thank you for your suggestion. We have revised our expression of MRI scan into scans. (line33)

2. all p-values need to be in small letters and in italics, or the authors need to be consistent in the way they report this.

Response: Thank you for your feedback. We will ensure that all p-values are presented in lowercase and in italics throughout the article to maintain consistency (See line 183, 193, 194, 201, 202, 204, 402, 406). It is important to note that in this article, many p-values are calculated using the permutation method, which has a specific formatting requirement for presenting these p-values. We referenced an article published in Nature Communications regarding the presentation of p-values in the permutation method as $P_{\text{permutation}}$.¹ Rest assured, we will make sure all p-values are consistently and appropriately reported. Your guidance is greatly appreciated.

3. Effect sizes need to be consistently reported please.

Response: Thank you very much for your suggestions. It is essential to consistently report effect sizes throughout the article. Please allow us to elucidate the statistical findings of this study: most of them involve the calculation of Pearson correlation values between RSFC and narrative indicators using cross-validation models, as well as the correlation between predictive indicators and actual indicators. We believe that the Pearson correlation coefficient itself already contains information about the strength and direction of the linear relationship between variables. In addition, indices such as the dice coefficient and the enrichment coefficient also appeared in this article. We used permutation method to calculate the significance of these values. Permutation methods are typically used to estimate the significance of observed sample statistics without assuming the overall distribution. By rearranging or reorganizing sample data, a random distribution based on the null hypothesis can be generated to evaluate the probability of observing the statistics under random conditions. If the observed statistics are very rare under the null hypothesis (i.e., a small p-value), the null hypothesis can be rejected, indicating statistical significance. Therefore, we hold the view that it is not necessarily required to calculate additional effect sizes, as the p-values provided by permutation method already reflect the characteristics of variable strength. Once again, Your meticulous standards and insightful guidance regarding this article are greatly appreciated!

1 Feng, J. *et al.* A cognitive neurogenetic approach to uncovering the structure of executive functions. *Nat Commun* **13**, 4588, doi:10.1038/s41467-022-32383-0 (2022).